# In Silico Study of Allosteric Communication Networks in GPCR Signaling Bias

**DOI:** 10.3390/ijms23147809

**Published:** 2022-07-15

**Authors:** Adrian Morales-Pastor, Francho Nerín-Fonz, David Aranda-García, Miguel Dieguez-Eceolaza, Brian Medel-Lacruz, Mariona Torrens-Fontanals, Alejandro Peralta-García, Jana Selent

**Affiliations:** Research Programme on Biomedical Informatics, Hospital del Mar Medical Research Institute, Department of Medicine and Life Sciences, Pompeu Fabra University, 08003 Barcelona, Spain; amorales@imim.es (A.M.-P.); francho.nerin01@estudiant.ufp.edu (F.N.-F.); david.aranda@upf.edu (D.A.-G.); miguel.dieguez01@estudiant.upf.edu (M.D.-E.); bmedel@imim.es (B.M.-L.); mariona.torrens@upf.edu (M.T.-F.); alejandro.peraltag@upf.edu (A.P.-G.)

**Keywords:** GPCR, signaling bias, functional selectivity, biased agonist, receptor dynamics, allosteric communication networks, molecular dynamics, network theory

## Abstract

Signaling bias is a promising characteristic of G protein-coupled receptors (GPCRs) as it provides the opportunity to develop more efficacious and safer drugs. This is because biased ligands can avoid the activation of pathways linked to side effects whilst still producing the desired therapeutic effect. In this respect, a deeper understanding of receptor dynamics and implicated allosteric communication networks in signaling bias can accelerate the research on novel biased drug candidates. In this review, we aim to provide an overview of computational methods and techniques for studying allosteric communication and signaling bias in GPCRs. This includes (i) the detection of allosteric communication networks and (ii) the application of network theory for extracting relevant information pipelines and highly communicated sites in GPCRs. We focus on the most recent research and highlight structural insights obtained based on the framework of allosteric communication networks and network theory for GPCR signaling bias.

## 1. Introduction

G protein-coupled receptors (GPCRs) are one of the largest families of cell surface receptors whose main function is to transduce extracellular stimuli into intracellular signals. They are involved in almost every process of human physiology and therefore represent a very important drug target. As a matter of fact, 42% of all FDA-approved drugs target at least one GPCR [1]. The observation that GPCRs can simultaneously trigger different signaling pathways creates novel opportunities to design drugs with an improved therapeutic effect [1]. In particular, molecules that are able to preferentially modulate disease-related pathways while sparing side effect-related pathways (so-called biased agonists) are promising drug candidates that can be superior to existing treatments. A better understanding of the molecular mechanisms that underlie the receptor function and signaling bias is therefore an important research focus for improved targeting strategies for a wide range of diseases.

Allostery and allosteric networks are elements critical for the function of this important drug target family. GPCRs are characterized by high inherent flexibility, which gives them the ability to sample a large conformational space [2,3]. Within this space, we find multiple inactive, intermediate, and active conformational states for specific pathways. The transition between them requires global structural rearrangements and correlated atomic movements that travel through intramolecular contact networks (so-called allosteric networks). Note that a GPCR adopts preferentially inactive conformational states in its basal state. However, due to its high flexibility, the receptor can, at times, be in active states, leading to basal signaling activities. The degree of basal signaling varies depending on the receptor type and environment.

Receptor activation is promoted by small molecules, peptides, or others (the so-called “agonists”) that bind to specific receptor pockets. There are two molecular mechanisms by which an agonist can mediate GPCR activation. On the one hand, an agonist can preferentially bind and stabilize active conformational states of the receptor, which shifts the equilibrium towards active states (active state sampling) [4]. On the other hand, agonist binding can induce perturbations and small structural changes that propagate across the receptor’s allosteric networks, which then leads to receptor activation (active state induction) [5]. Typically, both mechanisms can simultaneously contribute to receptor activation to different extents, depending on the ligand and receptor type [6]. Ultimately, active receptor states have favorable affinities to intracellular coupling partners, whose coupling eventually initiates specific downstream signaling cascades.

Importantly, within the framework of signaling bias, a biased agonist would shift the equilibrium of receptor conformations to states that allow selective interaction with one or a limited number of interaction partners, leading to a signaling bias. Equilibrium shifts between inactive, intermediate, and active receptor states are driven by the engagement of specific allosteric networks within GPCRs. Such allosteric networks are highly complex and often comprise thousands of intramolecular contacts. Small molecules (e.g., full or partial agonists) are able to differentially engage those networks, which promote different intracellular coupling and signaling profiles [7]. Unfortunately, the structural fundamentals and associated correlated movements of biased signaling are still poorly understood, as experimental approaches provide only partial insights. However, their broader characterization at a molecular level can improve the rational design of more effective and safer drugs. It is also relevant to mention that small variations within this network, as found in natural receptor variants or polymorphisms, can alter the response of GPCRs and can lead to disease conditions (e.g., retinitis pigmentosa, hyperthyroidism, diabetes, etc. [8]). If we understand the impact of such variations on the allosteric networks, we can develop strategies to rescue proper receptor function using small molecules.

In this review, we introduce the reader to the latest research on allosteric communication networks in the context of signaling bias in GPCRs. This includes computational approaches that can be used to detect complex allosteric networks in GPCRs. In addition, we outline the advances in the application of network theory methods to obtain an improved description of allosteric communication in GPCRs (e.g., relevant communication pathways and sites of high information flow).

## 2. Molecular Dynamics Simulations Provide Insights into the Dynamics of Allosteric Networks

In order to study the characteristics and functioning of allosteric networks in GPCRs, we require experimental setups that provide high-resolution insights. The most important approaches are X-ray crystallography and cryo-electron microscopy [3], which yield static information about atomistic contacts for most of the structural domains, with the exception of highly flexible regions (e.g., long extra- or intracellular loops (ECL or ICL)). These methods can be complemented with dynamics information using spectroscopy experiments, such as nuclear magnetic resonance (NMR), fluorescence resonance energy transfer (FRET), and bioluminescence (BRET) [9,10]. Typically, these methods provide local insights into receptor dynamics depending on the probe placement (e.g., outward movement of the transmembrane helix (TM) 6 upon receptor activation). In this scenario, molecular dynamics (MD) simulations are an excellent complementary approach to the aforementioned experimental setups, with the advantage of combining both high spatial and temporal resolution [6,11]. This technique allows us to simulate the structural motions of a protein in its natural environment. It uses classical physics to determine the forces that act on each atom of a molecular system and, consequently, their position after an interval of time. As a result, we obtain a high-resolution description of the trajectories of each atom of the protein and its environment over time. With these data, we can compute several metrics to understand how allosteric communication is mediated in GPCRs [6].

Classically, unbiased MD is suitable for sampling individual conformational states (i.e., active, intermediate, or inactive) within microseconds. However, the longer time scales required to observe receptor activation (i.e., hundreds of microseconds to milliseconds) are typically beyond the scope of this standard method. For this purpose, different computational approaches were designed in recent years to overcome this sampling problem. One of these techniques is accelerated MD, an enhanced simulation method that reduces the energetic barrier that has to be overcome to visit different receptor states [12,13]. This methodology has been used to study the activation of the M2 muscarinic receptor upon the binding of different ligands, yielding thermodynamic insights and detailed structural mechanisms based on the movement of residues [14,15]. Another technique is metadynamics, which discourages the exploration of previously adopted conformations, allowing for more efficient sampling of the conformational landscape than with standard MD. Recently, Saleh et al. investigated the complex formation of both the β-arrestin and G_s_-complexes with the β_2_-adrenergic receptor under the presence of a variety of ligands, which allowed them to characterize the cooperativity between a ligand and an intracellular binding partner [16]. Consequently, MD simulations and their different variations have proven to be useful tools in the study of allostery.

Most allostery studies have a common framework (Figure 1). In a first step, receptor dynamics data are generated via standard, coarse-grained, or enhanced simulation techniques. Then, we extracted/computed different metrics that were related to information transmission across the allosteric networks (e.g., inter-residue contacts and movement correlation). These metrics can be used to build network representations of the proteins and finally help to identify the main communication pathways or residues critical for allosteric communication.

## 3. Detection of Allosteric Networks in GPCR Signaling

Allosteric communication networks in GPCRs are highly complex and involve covalent and non-covalent interactions. Whereas covalent interactions are stable throughout the conformational sampling of different receptor states, non-covalent interactions are highly transient and their modulation is one of the main drivers for the functional response of the receptor. One prominent example is the disruption of the ionic lock at the intracellular site of the receptor, which involves a non-covalent interaction between an arginine in TM3 (R3.50) and a negatively charged residue in TM6 (D/E6.30)—first detected in bovine rhodopsin [18].

A non-covalent allosteric network in a prototypical class A GPCR is formed by thousands of atom–atom contacts between hundreds of amino acids, which makes its analysis challenging. A level of complexity is added by the fact that such non-covalent contacts are very versatile and include polar (hydrogen bonds), electrostatic (salt bridge and pi-cation), non-polar (hydrophobic and Van der Waals), stacking (pi stacking and T stacking), as well as indirect water-mediated interactions. The visualization of large interaction networks can be facilitated by flare plot representations (https://github.com/GPCRviz/flareplot, accessed on 11 July 2022) that depict interhelical connections between individual transmembrane domains TM1, TM2, etc., as provided by GPCRmd, an open-source and community-driven web resource that provides access to MD simulations of GPCRs [19] (www.gpcrmd.org, accessed on 11 July 2022) (Figure 2). Note that specific network types are spread differently throughout GPCRs. For instance, hydrogen bonding networks are largely distributed over the entire receptor, whereas salt bridge networks are significantly smaller. Importantly, during time-evolved network dynamics, we observe fluctuations in contact formation/disruption, which is measured as contact frequency (line thickness in flare plots, Figure 2), and provide valuable information about contact stability within the network. For instance, in the serotonin 5-HT_1B_ receptor, one of the most stable hydrogen bonds is found between the tyrosine Y7.43 and the well-known aspartate D3.32 (Figure 2A), a concerted interaction that is involved in ligand binding. On the other hand, we see a highly stable salt bridge in the 5-HT_1B_ receptor—the so-called ionic lock between R3.50 and E6.30—which keeps the receptor core at the intracellular site closed to stabilize the inactive receptor state (Figure 2B).

Most importantly, allosteric networks are modulated upon the binding of endogenous molecules or drugs (e.g., full, partial, biased agonists) to specific sites in a GPCR. Such ligand-binding sites can be considered important interfaces able to interfere (i.e., stabilize or perturb) with the inner receptor contact network to induce specific signaling responses. Hence, the evaluation of ligand–GPCR contacts has been an important focus in several studies of signaling bias. For instance, Stepniewski et al. described a common mechanism for ligand-binding contacts in aminergic receptors that are linked to (un)biased coupling responses [20]. According to their findings, ligands that establish simultaneous interactions with residues in TM5 and TM6 seem to have an unbiased coupling profile, whereas exclusive interaction with TM5 promotes G protein bias, in particular G_oB_. Note also that contacts with extracellular receptor regions (e.g., ECL2) drive signaling (un)bias, as reported for the Ghrelin receptor [21] or the adenosine 1A receptor [22]. Interestingly, in a later study, the authors propose that the same allosteric fragment (VCP171) binds to different allosteric sites in the ECL2 of the adenosine 1A receptor, which triggers different signaling responses. Mechanistically, ligand bindings and their specifically engaged contacts in orthosteric or allosteric sites modulate the inner receptor contact network, which stabilizes distinct conformational states that are linked to a specific un(biased) coupling response.

Despite their relevance for receptor function, relatively few computational studies have addressed ligand-engaged communication networks related to signaling responses. This is primarily due to their complexity comprising thousands of individual atom–atom contacts, making it highly challenging to interpret the results. In this scenario, an enhanced description of allosteric networks using network theory can be helpful in providing meaningful insights (see the section below).

In addition to contact interactions, there are several other useful structural descriptors that reflect the dynamics of allosteric communication networks in GPCRs (Figure 3). For instance, the root-mean-square deviation (RMSD) is a metric that can measure the average distance between selected atoms of a receptor structure compared to a reference one. Furthermore, when a receptor system fluctuates around a defined average position, the average RMSD over time is referred to as the root-mean-square fluctuation (RMSF). In other words, RMSF is a metric that reflects the atom mobility of each residue. In a study by van der Velden et al., the authors were able to find a correlation between the RMSF of the extracellular domain (ECD) and the (un)biased signaling profile for the GLP-1 receptor [23]. Therefore, the GLP-1 receptor showed markedly lower RMSF when bound to the endogenous GLP-1 peptide, which elicits an unbiased signaling response. In contrast, binding to a GLP-1 peptide variant (Ala8Val) resulted in a higher RMSF and a strong G protein biased over internalization.

Global motions of the receptor that occur at longer time scales are also tightly linked to the alteration of allosteric communication networks in the receptor. In this case, functional mode analysis (FMA) is a tool that identifies such primary movements in a receptor. This is achieved by projecting a principal component vector into one dimension. This process highlights the most important movements in a protein during the simulation. Using this analytical technique, Marmolejo-Valencia et al. studied the biased signaling mechanism of the μ-opioid receptor (μ-OR) [24]. The impact of biased agonists on information transmission in the μ-OR’s intracellular backbone was mainly observed in the helix 8 (H8) and ICL3 loop movements by FMA.

Finally, some studies performed analyses that focused more on pinpointing the precise coordination and dependencies that constitute the information transmission pathway. This also allowed for the identification of differences between biased and balanced ligands in terms of communication pathways and final structural and dynamic consequences. By this means, Suomivuori et al. elucidated the molecular mechanism of biased signaling on the Angiotensin receptor 1 (AT1R), which resulted in two major signaling conformations [25]. One conformation coupled primarily to β-arrestin, while the other coupled effectively to a G protein. The authors found that a long-range allosteric network involving residues along helixes 2, 3, and 7 allows ligands in the extracellular binding pocket to favor one intracellular conformation over the other. Using a similar approach, Cong et al. described differences in the dynamic and allosteric communications between the ligand-binding pocket and the receptor intracellular domains related to signaling bias in the μ-OR [26]. Interestingly, they observed that biased agonists trigger μ-OR conformational changes in the ICL1 and the H8, which may impair β-arrestin binding or signaling.

## 4. Enhanced Description of Allosteric Networks Using Network Theory

Network theory can aid in the analysis of allosteric communication in proteins. First, one must perform a bulk computation for residue-related parameters for the whole receptor from simulations such as interaction frequencies, movement correlations, or dihedral angle correlations. Then, each residue is represented by a node in a network, and the selected metrics are displayed as the weights of the edges connecting such nodes. From this network, it is possible to compute graph theory metrics to detect the main communication pathways between two parts of the protein or key residues for allosteric signal propagation. The frequently used network metrics are betweenness centrality, shortest pathways, and community analysis. For instance, the betweenness centrality analysis detects the influence of an individual node/connection on the information flow in a receptor network [27]. On the other hand, the shortest pathway is considered the biologically most relevant pathway [28]. Finally, community analysis allows us to detect sets of densely connected nodes. At the same time, it highlights which nodes are responsible for the communication between communities [29].

As an example, we have applied this concept to highlight differences in the communication network of the ergotamine-induced coupling profile for two serotonin receptors: 5-HT_1B_ and 5-HT_2B_. Ergotamine is an anti-migraine drug that elicits different coupling profiles at the 5-HT_1B_ receptor (activating G protein coupling and β-arrestin) compared to the 5-HT_2B_ receptor subtype (clearly favoring β-arrestin over G protein coupling) [30]. Computing movement correlations over all-atom MD data for both systems and applying a betweenness centrality analysis reveals regions with high relevance for the information flow that reaches from the receptor top down to the intracellular receptor side (Figure 4). We can appreciate clear differences between the 5-HT_1B_ (Figure 4A) and the 5-HT_2B_ (Figure 4B) receptors, which can be emphasized by plotting the delta network between both receptors (Figure 4C). Interestingly, according to this analysis, allosteric communication, particularly between TM6 and TM7, is altered overall by comparing unbiased (ergotamine induced 5-HT_1B_ receptor coupling response) and biased coupling conditions (ergotamine-induced 5-HT_2B_ receptor coupling response). Therefore, communication through the Y7.53 residue (part of the NPxxY motif) seems to have a special implication for discriminating between unbiased and biased signaling. 

In a different study, Bhattacharya et al. used the correlation in torsion angle movements combined with the shortest pathway algorithms to elucidate the allosteric pathways in different conformational states of the β_2_-adrenergic receptor: (i) the inverse-agonist-bound inactive state, (ii) the agonist-bound intermediate state, and (iii) the agonist- and G protein-bound fully active state [35]. They found that the inactive state shows dense clusters of allosteric pathways (allosteric pipelines) connecting the ECD with the intracellular domain (ICD). This results in reduced receptor dynamics compared to the intermediate and active states. Interestingly, these pipelines are weakened in the intermediate state due to the decoupling of the ECD from the ICD, resulting in a more dynamic receptor. The study of allosteric pipelines between the ligand-binding pocket and the ICD has also been applied to obtain a better understanding of biased agonism at the β_2_-adrenergic receptor, serotonin 5-HT_1B_ and 5-HT_2B_ receptors and for G protein–biased agonists in the κ-opioid receptor [36]. Therefore, the ratio between pipelines related to G protein or β-arrestin binding provides an estimate for the degree of bias for the ligand–receptor complex.

In a similar study on the β_2_-adrenergic receptor, Chen et al. focused on the molecular mechanism of different biased agonists [28]. For this, the authors computed movement correlation networks combined with the shortest pathway analysis between the binding pocket and the intracellular coupling domain. Interestingly, the shortest detected pathways highlight residues that are known to take an important role in bias and receptor function. The agreement between these methods and previous experimental observations reinforces their utility in analyzing information transfer in proteins.

Furthermore, a long-standing challenge in MD has been to observe rare events that are related to the functional response of a GPCR. Plante et al. used unsupervised machine learning to detect such rare events in the dynamic contact network of the 5-HT_2A_ receptor [37], comparing the functional impact of its native agonist, 5-HT, and the selective inverse agonist ketanserin. Their analysis method consisted of applying non-negative matrix factorization to the matrix representation of the dynamic contact network. This process highlights key events that take place over the simulation while ignoring random noise in the contact network variations. For instance, the authors find that the serotonin-bound 5-HT_2A_ receptor undergoes a structural rearrangement in the ICL2, which induces an increase in the volume of the intracellular cavity—an activation-related event that is observed in crystal structures when bound to the heterotrimeric G protein [37].

Overall, graph-aided analysis of allosteric communication networks can be a useful approach for obtaining deeper insights into the molecular mechanisms of receptor function. The advantage is that this methodology allows for high throughput analyses that are able to implement a wide variability of user-defined structural descriptors. However, despite its potential, future work is required to further improve the application and data interpretation of the graph-aided analysis of allosteric communication networks for GPCR research.

## 5. Conclusions

In this review, we have introduced the reader to the computational study of allosteric communication networks in the context of GPCR functionality and signaling bias. Allosteric communication networks are highly complex, and we are only starting to understand their dynamics and implication in the receptor response. Classical studies often use simplified structural descriptors (e.g., inter-residue contacts and dihedral angles) to monitor consequential events that link the ligand binding site to the intracellular coupling site of effector proteins. An enhanced description of allosteric communication networks within the entire protein using network theory and machine learning approaches can provide a more complete picture. By using these approaches, several studies have been able to detect long-range allosteric networks that induce an alteration in the intracellular coupling site of GPCRs. In turn, this favors the binding of only one specific coupling partner and therefore promotes signaling bias.

Currently, we are experiencing a very exciting era because of the increasing structural knowledge of GPCRs. Combined with powerful computational approaches, we can expect that in the future, this will significantly improve our understanding of the connection between the information flow through allosteric communication networks and the final receptor signaling response.

## Figures and Tables

**Figure 1 ijms-23-07809-f001:**
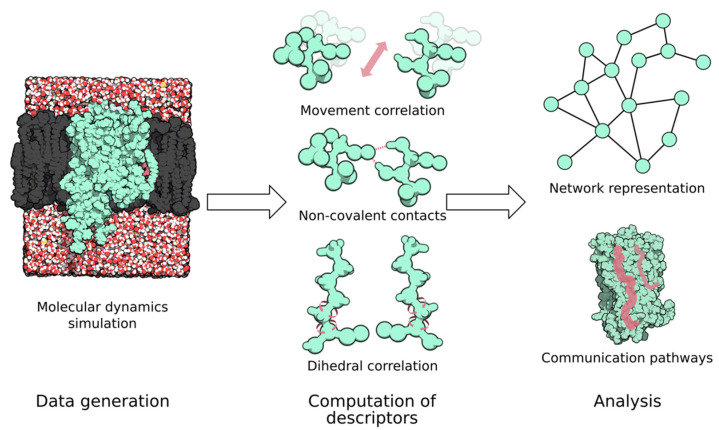
The common framework for the detection of allosteric networks comprises data generation, computation of descriptors (e.g., movement correlation, intramolecular contacts, etc.), and network representation and analysis. Molecules have been rendered using the QuteMol software (v 0.4.1; Marco Tarini, Paolo Cignoni; Pisa, Italy) [17].

**Figure 2 ijms-23-07809-f002:**
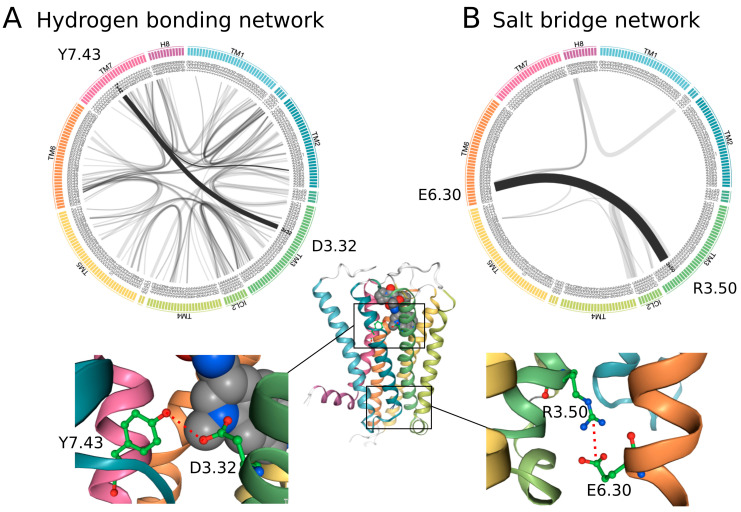
Allosteric communication networks for the serotonin 5-HT_1B_ receptor on the online GPCRmd resource for (**A**) the hydrogen bonding network and (**B**) the salt bridge network. Flare plots allow for a concise depiction of interhelical contacts between the transmembrane helix (TM)1, TM2, etc. The thickness of the TM domain-connecting lines reflects the contact frequency computed over time-evolved network dynamics. The interaction with the highest contact frequency for the hydrogen bonding and salt bridge network is highlighted.

**Figure 3 ijms-23-07809-f003:**
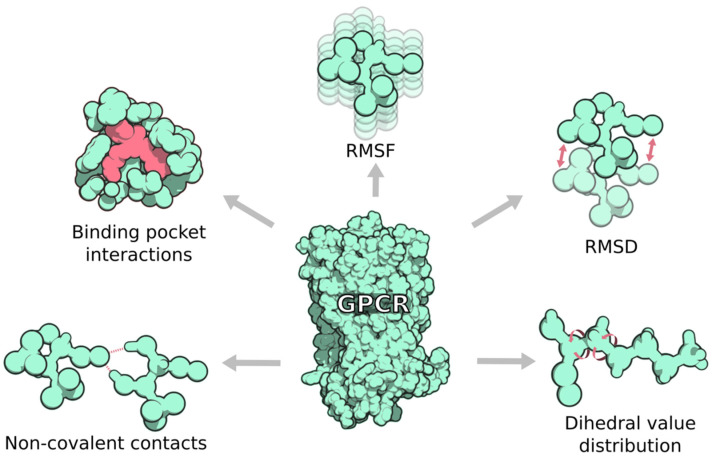
Common metrics for allosteric communication analysis. (i) Non-covalent contacts are very versatile and comprise hydrogen bonds, van der Waals interactions, etc. They can be computed within the protein or between the protein and its ligand. (ii) Root-mean-square fluctuations (RMSF) can be used to compute the effects of allosteric communication in protein mobility. (iii) Root-mean-square deviation (RMSD) is used to compute differences between conformations of the same system. (iv) Distributions of dihedral rotamers in the protein can inform about changes in the allosteric communication networks. Molecules have been rendered using the QuteMol software [17].

**Figure 4 ijms-23-07809-f004:**
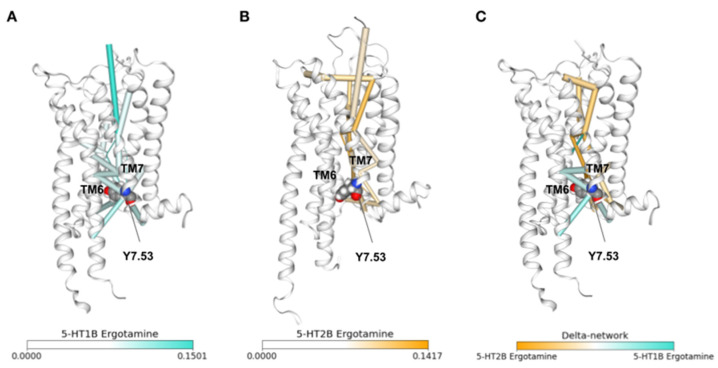
Allosteric communication networks of the (**A**) serotonin 5-HT_1B_ and (**B**) 5-HT_2B_ receptors bound to ergotamine (a balanced and β-arrestin biased ligand) and (**C**) their delta-network. structures and molecular dynamics (MD) simulations are taken from the GPCRmd database IDs 90 and 94, respectively. Allosteric communication network edges were computed with dynetan [31]. The subset of edges between residue pairs that were in contact (according to GetContacts (https://getcontacts.github.io/, accessed on 11 July 2022) contact frequencies) and more than five positions away in the sequence were used for edge betweenness centrality analysis with NetworkX [32], which are the values used for representation. The 20 edges with the highest values are represented in each case, colored according to the corresponding color bar, and Tyr 7.53 from the NPxxY conserved motif is shown with spheres in the structures. The delta network of the two structures highlights the edges that are the most different (i.e., have the most different values) between the two individual networks and is calculated by subtracting the edge values of one network from the other. The corresponding residues between the two structures for the subtraction of edge values are those provided by a structural alignment of the initial PDBs performed with T-Coffee [33], using the TM-align pairwise structural alignment method [34].

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
