# Peer review of "In Silico Study of Allosteric Communication Networks in GPCR Signaling Bias"

_ijms, 2022, doi:10.3390/ijms23147809_

Round 1

Reviewer 1 Report

The authors summarized the latest studies on the computational approach for allosteric communication and biased signaling in GPCR. Overall, this is an informative and useful review to broad readers interested in GPCR. 

I have a few relatively minor suggestions explained below. 

1. What "GoB" in line183 means? Is that just a typo? If not, the author should explain more detail.

2. RMSD is mentioned only in the legend of Fig3. If it were explained in the main text, it could be more informative to readers.

3. In line 263,264, the authors explained the biased agonism of Ergotamine to 5-HT1B and 2B. The references for this biased agonism are absent. The author should include the references.

Author Response

Dear Reviewer, 

We appreciate the time and effort that you have dedicated to providing your valuable feedback on our manuscript. We have incorporated all your suggestions.

Here is a point-by-point response to your comments.

  • Comment 1: What "GoB" in line 183 means? Is that just a typo? If not, the author should explain more detail.

Response: Thank you for pointing this out. GoB corresponds to the G protein alpha subunit subtype “oB”. We have subscripted the subtype “oB” to avoid misunderstandings. We have performed the same operation with Gs, which was mentioned also in line 130. 

  • Comment 2: RMSD is mentioned only in the legend of Fig3. If it were explained in the main text, it could be more informative to readers.

Response: We agree with the reviewer. We have, accordingly, included a description of RMSD in lines 121-125.

  • Comment 3: In line 263,264, the authors explained the biased agonism of Ergotamine to 5-HT1B and 2B. The references for this biased agonism are absent. The author should include the references.

Response: Thank you for noticing this. We have included the corresponding reference in the manuscript as reference 30. The corresponding citation in the bibliography section is the following. 

Wacker, D.; Wang, C.; Katritch, V.; Han, G.W.; Huang, X.-P.; Vardy, E.; McCorvy, J.D.; Jiang, Y.; Chu, M.; Siu, F.Y.; et al. Structural Features for Functional Selectivity at Serotonin Receptors. Science 2013, 340, 615–619, doi:10.1126/science.1232808.

We hope that with these corrections we satisfy your expectations. Thank you again for your time and effort.

Reviewer 2 Report

The authors have undertaken a fairly comprehensive review of computational methods and techniques for studying allosteric communication and signaling bias in G protein-coupled receptors.  I did not have any concern in this manuscript.

Author Response

Dear Reviewer, 

We appreciate the time and effort that you have dedicated to reading and evaluating our manuscript. We are grateful for your positive feedback and for having met your quality expectations.